# The Perspective of Rehabilitation Nurses on Physical Exercise in the Rehabilitation of Older People in the Community: A Qualitative Study

**DOI:** 10.3390/jfmk8040163

**Published:** 2023-11-28

**Authors:** Rogério Ferreira, Nuno Fernandes, Carina Bico, Ana Bonito, Cláudia Moura, Luís Sousa, Cristina Lavareda Baixinho, César Fonseca

**Affiliations:** 1Polytechnic Institute of Beja, School of Health, 7800-111 Beja, Portugal; 2Comprehensive Health Research Centre, 7000-811 Évora, Portugal; 3Unidade Local de Saúde do Norte Alentejano (ULSNA), Portalegre Hospital, 7300-853 Portalegre, Portugal; n_alexfernandes@hotmail.com (N.F.); karinaamieira@gmail.com (C.B.); abitinha@gmail.com (A.B.); luismmsousa@gmail.com (L.S.); cfonseca@uevora.pt (C.F.); 4Setúbal Hospital Center, S. Bernardo Hospital, 2900-182 Setubal, Portugal; claudiabmoura@sapo.pt; 5School of Health, Atlantic University, 2730-036 Barcarena, Portugal; 6Lisbon School of Nursing, 1900-160 Lisbon, Portugal; 7Nursing Research, Innovation and Development Centre of Lisbon (CIDNUR), 1900-160 Lisbon, Portugal; 8Escola Superior de Enfermagem S. João de Deus, Universidade de Évora, 7004-516 Évora, Portugal

**Keywords:** aged, rehabilitation nursing, exercise, healthy aging, quality of life

## Abstract

In the face of an increasingly aging population, nurses have to design and implement programs aimed at the elderly in order to keep them healthy and maintain their quality of life. Structured physical exercise and overall physical activity play a major role in maintaining an active lifestyle, improving health, preventing disease, and helping to maintain older people’s quality of life. To investigate the importance of implementing physical exercise programs for the older person in the community, taking into account the perspective of the rehabilitation nurse specialist, this is a qualitative, exploratory, and descriptive study with nine rehabilitation nurses from various regions of Portugal who have experience of implementing physical exercise programs with the elderly. This study used semi-structured interviews, one of the most common data collection procedures in social and health research. Content analysis was used to analyze the data. The study received a positive opinion from the Ethics Committee. The participants’ perspectives focused on the following subjects: “Physical exercise program for the elderly in the community”, “Importance of physical exercise in the rehabilitation of the elderly in the community”, “People’s adherence to the programs”, “Gains in health resulting from the implementation of these strategies” and “Gains from group activities”. A set of indicators emerged from the analysis. Nurses recognize the importance of using structured physical exercise programs adjusted to the rehabilitation of the older person, with gains in promoting active and healthy aging.

## 1. Introduction

The world’s population is aging and every country in the world is seeing an increase in the number and proportion of older people in its population. Population aging is set to become one of the most significant social transformations of the 21st century. It is estimated that the number of older people aged 60 and over will double by 2050 and triple by 2100, rising from 962 million in 2017 to 2.1 billion in 2050 and 3.1 billion in 2100 [1].

As in other European countries, Portugal is no exception, with an uninterrupted increase in the elderly population and a decrease in the young population in recent decades [2]. In 2021, the percentage of the population aged 65 or over represents 23.4%, while the percentage of the population under the age of 14 is 12.9%, i.e., the increase in longevity and the decrease in the birth rate which have taken place in recent years are reflected in the age pyramid in the censuses from 2011 to 2021, which show a narrowing at the bottom and a widening at the top where the most advanced ages are found [2]. In 2021, the aging index showed 182 older people for every 100 young people, with the average age in Portugal being 45.4 years [2].

In geographical terms, once again taking into account the data provided by the INE [2], the increase in the aging index was common to all geographical areas; however, the Central and Alentejo regions had the highest aging index, with 229 older people per 100 young people in the center and 219 older people per 100 young people in Alentejo.

Ensuring healthy aging and quality of life for the older person is an emerging challenge in today’s society, where longevity is increasing in parallel with an increase in the degree of total dependence of citizens. This situation has implications for healthcare, research, and health policies [3].

The challenges of maintaining health, well-being, and other relevant aspects linked to the quality of life of older people living in the community or institutionalized have been the subject of multiple studies.

In recent decades, it has been shown that structured physical exercise and overall physical activity play a major role in maintaining an active lifestyle, improving health, preventing diseases, and helping to maintain older people’s quality of life [3,4]. They work as preventive strategies for many chronic diseases, improving mobility and mental health, reducing mortality, and many other benefits [4].

Along these lines, it is essential to differentiate between the terms “physical activity” and “physical exercise”. Physical activity refers to any bodily movement produced by the skeletal muscles which leads to an increase in energy expenditure. Its intensity and duration can vary substantially. Physical exercise is a subcategory of physical activity, which is assumed to be planned, structured, and repetitive, and in which movements are performed with or without the explicit intention of maintaining or improving one or more components of physical fitness, namely aerobic capacity, muscle strength, flexibility, balance, and motor coordination [4].

The design and implementation of physical exercise programs must comply with the following parameters: intensity, frequency, time (duration), type of exercise, volume, and progression of the exercise program [5], depending on the older person’s health situation, their motivation and the context in which the training program is carried out. In conceptual terms, intensity is related to carrying out the exercise program safely. It can be determined by objective parameters such as heart rate and oxygen consumption and by the subjective perception of effort as a subjective parameter [5]. Frequency refers to the number of times the exercise program is carried out per day and/or per week, while time refers to the duration of each session. Typology corresponds to the type of exercise program carried out and volume is related to the total amount of exercise carried out, considering the time of the exercise program, the speed, the number of repetitions and the load. Progression has to do with how the programmed training volume is achieved [5].

These rehabilitation strategies for the older person, involving structured physical exercise programs, are crucial for active and healthy aging, which ensures the delay of functional losses and the maintenance and development of the functional capacity of this population [6].

This study focuses on the implementation of structured physical exercise programs for older people in the community. Several studies have focused on the positive effects of these programs on older people, whether in a community setting or in residential facilities for older people [7,8,9,10,11,12,13,14,15,16,17].

Given the problems that many older people experience as the years go by, it is necessary for rehabilitation nurses, in conjunction with other partners in the community, to be able to design and implement physical exercise programs, and other strategies to promote active aging, which guarantee the support of older people in the community, the promotion of social participation, and the promotion of healthy lifestyles [18].

In this study, the researchers considered it pertinent to analyze the perspectives of nurses specializing in rehabilitation nursing on physical exercise programs for the older person in the community, and we considered the following specific objectives:–To understand the importance of physical exercise in the rehabilitation of the older person in the community.–To characterize the physical exercise program implemented for the older person in the community.–To identify the health gains resulting from the implementation of these strategies.

The results of this study can make important contributions to reflections on this issue and to the definition of health policies which are helping in the valorization and integration of physical exercise programs in the health care of elderly people in the community. We believe that these contributions may highlight the relevance of specialized rehabilitation nursing interventions in the implementation of strategies which promote active and healthy aging in older people [3,4].

## 2. Materials and Methods

### 2.1. Study Design

This qualitative, exploratory, and descriptive study is based on the care processes in which nurses have intervened in the community and aims to understand their perspectives on physical exercise in the rehabilitation of the older person.

### 2.2. Setting and Participants

The participants are nurses with specialized training in rehabilitation nursing who provide nursing care in the community in various locations in Portugal. An intentional (nonprobabilistic) sample of nine rehabilitation nursing specialists was selected, with a wealth of experience in developing physical exercise programs for the older person in a community setting. The inclusion criteria were to have specialized training in rehabilitation nursing and to have participated in physical exercise programs with older people in the community. No exclusion criteria were defined. The number of participants in the sample was determined by the criterion of saturation of the findings.

### 2.3. Data Collection

Individual interviews were used, being one of the most widely used data collection procedures in social research, education, and health, particularly when it involves qualitative methodology. A semi-structured interview script was developed and submitted to two judges to ensure the quality of the measuring instrument [19]. This was a process whose main purpose was to ensure that the questions asked were in line with the central objective of the study and that they showed clear language and relevance to the field under study [20].

The first part of the interview script elicits data which ensure the legitimacy of the interview. In order to understand the nurses’ perspective on physical exercise in the rehabilitation of older people in the community, the following questions were asked:–What did the physical exercise program for older people in the community consist of?–What components do you take into account when planning physical exercise with the older person?–How did the physical exercise program contribute to the rehabilitation of the older person?–How do you analyze people’s adherence to these programs?–What health gains have resulted from implementing these strategies?–Do you think there are any advantages to carrying out these programs in groups? Can you name some of them?

Contact with the participants was made through one of the researchers and included a presentation of the research project and an explanation of the study’s purpose, objectives, and the importance of their participation. This was conducted by telephone and email.

The interview was conducted by one of the researchers via videoconference using the Zoom^®^ platform (San Jose, CA, USA), on dates and at times agreed with the different participants. The interviews lasted approximately 90 min, at a time agreed with the participants. They were recorded and later transcribed into a Word file, with the participants’ permission.

The participants approached the topic freely, focusing on their experiences of caring for the elderly in the community, with the help of the interview script.

### 2.4. Data Processing and Analysis

To analyze the data, the content analysis technique was used, more specifically thematic categorical analysis, which made it possible to discover the core meanings of communication [21]. To implement this technique, it was necessary to take into account the following steps:Definition of objectives and a guiding frame of reference;Constitution of a corpus;Definition of categories;Definition of analysis units.

The definitions of objectives and a reference framework provided structure for data collection and analysis. The full transcriptions of the nine interviews became the corpus of the analysis, corresponding to the material to be analyzed and produced for the research, in line with Bardin’s [21] recommendations.

Categories are headings or classes which bring together a group of elements (registration units), under a generic title [21]. This term indicates the central meaning of the concept. The construction of the system of categories can be based on the objectives of the study (a priori) or from what emerges in the participants’ speech (a posteriori). In this study, this process predominantly resulted from the analysis of the participants’ speech, and the conceptual title of each category was defined during the analogical and progressive classification of recording units (themes), taking into account that it was an open and inductive procedure [21].

The definition of analysis units structures data coding and comprises recording units and context units. For Bardin [21], the registration unit is the unit of meaning to be encoded and corresponds to the content segment to be considered as the base unit. In a thematic categorical analysis, the basic unit is the theme, which is understood as a statement about a subject. It can be a sentence or a compound sentence, under the influence of which a vast set of singular formulations can be affected [21]. The context unit was considered as the understanding unit to encode and understand the exact meaning of each recording unit (theme). In this study, we defined each participant’s response to the questions asked as a context unit.

Thus, in the first phase of the thematic categorical analysis, a “floating reading” was carried out, with the aim of verifying whether the information collected was related to the objective of the study. Next, it was necessary to classify the different recording units (themes) into different categories, according to criteria capable of giving meaning to the analysis [21]:iMutual exclusion: each registration unit (theme) cannot exist in more than one category;iiExhaustiveness: the entire text, which forms part of the corpus, must be analyzed;iiiHomogeneity: the same set of categories can only operate with one dimension of analysis, which follows the same principles;ivObjectivity: different parts of the same material must be coded in the same way, that is, different coders must reach the same results;vRelevance: categories are considered relevant when they are adapted to the objectives and the defined reference framework.

Compliance with these criteria made it possible to implement what is implicit in this content analysis technique, thematic categorical analysis, which works through operations of breaking the text into units and categories according to analogical regroupings. It is about dismembering a discourse and producing a new one, given the discovery of the relationships between the parts of that discourse [21].

The development of a thematic categorical analysis presupposes the discovery of the nuclei of meaning that make up communication and in which their presence or frequency of appearance mean something in relation to the objective of the analysis [21]. This means that there are enumeration rules which the researcher must define to structure his analysis. The presence of indicators or their frequency are two of the most used criteria. In this study, we considered the presence of this indicator as the core meaning of this communication, a rule which fits with what characterizes qualitative analysis [21].

### 2.5. Rigor and Trustworthiness of the Research

To ensure the rigor and reliability of the investigation and in line with the recommendations of Velloso and Tizzoni [22], several procedures were developed. The researchers discussed the different decisions until consensus was reached during the analysis process and returned the analysis results to the participants to confirm the researchers’ interpretations. These procedures were essential to improve the study’s credibility. Furthermore, quotations from the participants’ speech were presented to ensure that those seeking to transfer the results to their location could assess transferability. Dependability/reliability is a criterion which focuses on the ability of external researchers to follow the methods used by the researchers and which was ensured through the detailed description of each stage of the decision-making process, ensuring that readers could follow the research. Confirmability is the ability of other external researchers (judges/experts) to confirm the researchers’ constructions and to guarantee this criterion of rigor. Two judges with experience in the data analysis technique and knowledge in the field under study were asked to look for discrepancies, comparing their perceptions with those of the researchers, and comparing their analyses of the interviews with those of the researchers. In this case, we sought to verify the researchers’ interpretations, making sure that they were not figments of their imagination.

### 2.6. Ethical Aspects

All necessary efforts were made along with the institutions and study participants to respect the ethical principles that should guide the preparation of a study such as this. At an institutional level, the study under reference SC 2023/1695, referring to request no. 02/2023 dated 8 February 2023, received a positive opinion on 27 February 2023 from the Ethics Committee of the Polytechnic Institute of Portalegre. Participants were assured that their participation was voluntary and that they could withdraw from the study at any time. They were guaranteed anonymity and the confidentiality of their data, being reassured that professional discretion was an obligation and duty. The participants also signed a free and informed consent term.

The researchers took responsibility for managing the data collected, ensuring anonymity and confidentiality. Information which could identify the participants was restricted by assigning a unique code to each participant. Participants’ names were replaced by identification numbers (P1, P2, P3, …) in the records and publications. In the recording units which were presented in the study results, measures were taken to protect the identity of the participants. Data protection extended from the selection of participants to the collection, analysis, and dissemination of study results.

The information collected and submitted for analysis, in addition to the informed consent documents, was stored on the research team’s own hard drive for a minimum guaranteed period of 5 (five) years. The data were encrypted, including the raw data, to prevent unauthorized access.

## 3. Results

This study involved nine nurses with postgraduate training in nursing. All the nurses had the title of specialist nurse in one of the areas defined by the Portuguese Nurses’ Association, the area of rehabilitation nursing. In this sample, five nurses are male and four female. Their length of service as nurses varies between 14 and 29 years, with an average of 19.2 years. They have specialized training in rehabilitation nursing for between four and sixteen years, with an average of 9.7 years. The nine participants in this study work in the regions of Lisbon (3), Porto (2), Coimbra (1), Vila Real (1), Santarém (1) and Acores, in S. Miguel (1).

The content analysis process involved breaking down the text into registration units (themes) and categories, according to analogical regrouping, as advocated by Bardin [21]. A semantic categorization criterion was used for this purpose.

This study involves five categories: “Physical exercise program for the elderly in the community”, “Importance of physical exercise in the rehabilitation of the elderly in the community”, “People’s adherence to the programs”, “Health gains resulting from the implementation of these strategies” and “Gains from group activities”. Several indicators emerged from the participants’ discourse and as a result of the analogical and progressive classification of the recording units, as shown in Table 1.

### 3.1. Physical Exercise Program for the Older Person in the Community

Two subcategories are considered in this category, Purpose and assessment and Components.

#### 3.1.1. Purpose and Assessment

The Purpose and assessment subcategory includes the indicators Program, Program objectives, Person assessment, Assessment tools, and Conditioning factors.


**Program Definition**


The physical exercise program is aimed at the day-care population and is structured according to the FITT model. These characteristics are highlighted by two rehabilitation nurses.

The program consists of implementing a physical exercise promotion program for the day-care population in our catchment area.(P2)

Subsequently, a training plan/program is designed according to the FITT model (Frequency, Intensity, Type, Time), thus personalizing each program.(P7)


**Program objective**


The program involves motor and respiratory re-education. It is decisive in the performance of activities of daily living and in the elderly person’s quality of life. These aspects are highlighted in the following two statements.

Physical exercise improves muscle function, strength, and endurance as well as the respiratory muscles, thereby reducing dyspnea in people with respiratory disease.(P6)

In this line of thought, the aim will be to plan an exercise training program that allows the elderly person to reduce symptoms, improve their ADL performance, and increase their quality of life.(P6)

“a Respiratory Rehabilitation Program for patients with COPD…”(P9)


**Person assessment**


The program has an initial assessment, a mid-point assessment, and a final assessment which is decisive in defining the program. It involves assessing muscle strength, joint range of motion, flexibility, balance, functional capacity, cardiovascular risk, and lifestyle habits. The following statements highlight these aspects.

(This program has) (…) an initial, intermediate, and final assessment.”(P8)

“…which involved an initial individual assessment of all users…”(P9)

When I plan my sessions, I base them on a full assessment. It’s the assessment that will determine the type of exercises chosen. In this initial assessment, which is also intermediate and final, I assess strength, balance, joint range of motion, flexibility, and cardiovascular fitness (…). I also assess cardiovascular risk and lifestyle habits, including diet, rest, and medication.(P8)

A multidimensional assessment of the elderly person is carried out and the altered functional dimensions that are sensitive to specific RNS interventions are identified, using the Clinical Functional Vulnerability Index.(P1)

The physical exercise program offered to the elderly is based on a first approach to **assess** their capacity in terms of physical condition.(P7)

The professionals at the day center monitor blood pressure and heart rate on a weekly basis.(P2)

(…) assessment of clinical risk of developing cardiorespiratory complications and estimation of maximum heart rate on exertion (…)(P3)

The assessment also makes it possible to identify the objectives that the person wants to achieve with the program and the organization of the training, in terms of intensity.

It is important to carry out a subjective evaluation centered on the person: “What would be important to you?” (before the program), “Which of your objectives were achieved with the program?” (after).(P6)

(…) before being exposed to the group, the ability to assess each person must be developed in order to avoid embarrassment when practicing physical exercise. Then, people who are as similar as possible in terms of performance should be chosen, so that the intensity of the training can match everyone in a longitudinal way.(P7)


**Evaluation tools**


The instruments used focus on assessing respiratory and motor functions, as well as the older person’s quality of life. These aspects are evident in the following statements.

An initial assessment is carried out on the elderly participants, using an evaluation questionnaire and scales to monitor their respiratory and motor function and quality of life: Dyspnea, Modified Medical Research Council (mMRC), 1-min sit and stand test, World Health Organization’s Abbreviated Quality of Life Assessment Tool (WHOQOL—BREF).(P2)

(…) the 6-min walk test.(P8)


**Conditioning factors**


The underlying pathology, the preferences of the elderly person, and the objectives set, as well as the physical space where the program takes place are conditioning factors, expressed in the following recording unit.

The underlying pathology that determined the need to implement the training program, the person’s preferences in terms of exercises, the space where the session will take place, and the objectives set (increasing aerobic capacity, muscle strength, ADL training, among others).(P5)

#### 3.1.2. Components

The subcategory ‘components’ includes the indicators program intensity, frequency and time, type of program, progression, and educational component.


**Program intensity**


The intensity of the program is usually determined by applying the modified Borg scale. On the other hand, exercise tolerance must be taken into account. When the conditions exist, the intensity of the program should be increased in order to improve the elderly person’s physical condition.

Intensity—Use of the modified Borg scale with a max of 6 or application of the Karvonen formula for people at cardiorespiratory risk.(P3)

Intensity—Use of the modified Borg scale, where a score of 4 to 6 for dyspnea and fatigue indicates high intensity. For the elderly, low-intensity training can be considered, considering the objectives described above.(P6)

Exercise tolerance: interval training should be considered.(P6)

This component of increasing the intensity (…), exercise difficulty is very important, because it is only this increase in exercise difficulty that leads to an improvement in the elderly person’s physical condition.(P8)


**Frequency and Time**


The physical exercise programs run by rehabilitation nurses vary in terms of frequency and time. The predominant frequency is two to three times a week and approximately 1 h. The length varies between 16 and 20 sessions, although in one case the program lasted 6 months.

Physical exercise sessions twice a week, duration 1 h.(P1)

(…) program of 16 sessions, 8 weeks, two sessions per week of 1:30 h (in hospital gym).(P4)

(…) 20-session program, 7 weeks, three sessions per week lasting 1 h (via tele-rehabilitation).(P4)

These programs last 6 months (…). They involve exercising three times a week for 1 h each session.(P8)

Frequency—strength training two to three times a week (initially supervised), aerobic training five times a week.(P3)

Frequency—two to three times a week (…)(P6)

Time—30-min sessions, which can increase to 45 min.(P3)

Duration: minimum 20 sessions, minimum 20 min.(P6)


**Type of program**


Physical exercise programs are varied. They include warm-up exercises, program-specific exercises, and recovery or relaxation exercises. The exercises in the program should be tailored to the older person and aim to strengthen muscles, and improve joint mobility, flexibility, balance, and the cardiorespiratory aspect. There are also programs that include aerobic training.

The physical exercise program comprises several phases, warm-up, exercise component (which includes muscle strengthening exercises, exercises aimed at working on the cardiorespiratory system, exercises to promote proprioception and balance), and the recovery phase.(P8)

Warm-up exercises (5 min., once a week, 1 year), Muscle strength and balance exercises (20 min., once a week, 1 year), Flexibility exercises (20 min., once a week, 1 year), Relaxation exercises (5 min., once a week, 1 year).(P2)

Types of exercises—active, balance, joint mobility, motor coordination, and flexibility. Usually no aerobic exercises.(P1)

Type—Resistance and/or strength training (…)(P6)

Type—Aerobic exercises: walking, step, bicycle/cyclo ergometer; strength exercises: squats; push-ups, use of dumbbells, shin guards for large muscle groups, abdominals, lower back and upper back using elastic bands.(P3)

It essentially consists of aerobic training to improve aerobic capacity, with a focus on gait training, which can be carried out on an electric treadmill or even outside the home; as well as strength training using dumbbells, elastic bands, and body weight. Finally, balance and coordination training is widely used.(P5)

Exercise training—warm-up, aerobic training, strength training, balance training (if necessary), and stretching.(P4)

The program is tailored to each individual and takes various aspects into account. Telerehabilitation is also used to carry out exercise training in one of the situations.

A physical exercise program should be tailored to each person, regardless of age.(P6)

The components to be taken into account are varied and fall into multiple domains, and in our service, in addition to the cardiorespiratory and cognitive components, functionality, autonomy and quality of life must always be taken into account.(P7)

We currently have two outpatient respiratory rehabilitation programs underway, in a hospital gym (1) and telerehabilitation (2), where exercise training is one of the fundamental pillars.(P4)


**Volume and Progression**


Volume is related to the number of repetitions. Progression is essentially focused on the load (volume). Repetitions vary between eight and 12 and progression is determined by the effort in the last two repetitions, as expressed in the following recording units.

Progression—Measure the load at which the person can do 10 repetitions and go from there. Progress in repetitions or resistance as the person stops doing the last two repetitions in effort. Between eight and 12 repetitions in strength exercises.(P3)

Progression: Progression in time and/or load (…)(P6)


**Educational component**


Education should be one of the components of the program, as expressed by this participant.

Educational component—all areas covered in the Living well with COPD manual.(P4)

### 3.2. The Importance of Physical Exercise in the Rehabilitation of the Older Person in the Community

This category includes the indicators “Health promotion”, “Functional capacity” and “Well-being and self-care”.


**Health promotion**


Physical exercise programs are important for health promotion, promoting healthy lifestyle habits, and reducing the risk and incidence of falls.

Promotes healthy lifestyle habits (…)(P2)

(…) reduces the incidence of falls (…)(P2)

(…) reduces the risk of falls (…)(P3)


**Functional capacity**


Physical exercise is important for improving older people’s functionality. The impact of these physical exercise programs on functionality and on cognitive function has repercussions on the elderly person’s performance and activities of daily living (basic activities of daily living and instrumental activities of daily living). The following statements express this importance.

Improved physical and cognitive performance, with an impact on their activities of daily living and instrumental life activities.(P1)

The current programs allow for an increase in muscle strength, endurance, and power, improved flexibility and balance, improved functionality, improved cardiovascular health, and reduced perception of dyspnea and fatigue (…)(P4)

The exercise program enhances the physical, motor, and balance capacity of the elderly, which allows them to increase their functionality (…)(P8)


**Well-being and self-care**


Physical exercise programs are a determining factor in the elderly person’s sense of well-being, quality of life, and self-care, as expressed in the following recording units.

(…) improved symptoms of anxiety and depression, improved performance in activities of daily living, and improved health-related quality of life.(P4)

It has improved biomechanical performance, making it possible to carry out activities of daily living more independently; it has increased aerobic capacity and tolerance to exertion; and it has increased muscle strength, allowing the person to carry out activities that they were previously unable to do, such as cooking, going to the toilet independently, carrying light weights such as shopping and various objects.(P5)

Through physical exercise programs, the elderly person will improve muscle fatigue, dyspnea, and other symptoms, certainly contributing to a better quality of life. It also prevents premature immobility and promotes a more active life, contributing to their physical, emotional, and family well-being.(P6)

(…) and the person is better able to carry out activities of daily living.(P8)

… it has contributed to an improvement in symptoms and a slower progression of COPD, and the improvement in mental health, namely the risk of depression and anxiety, is quite evident.(P9)

### 3.3. People’s Adherence to Physical Exercise Programs

This category includes the indicators “Feedback and satisfaction” and “Involvement”.


**Feedback and satisfaction**


Adherence was expressed by feedback or expressions of satisfaction by older people about the programs or the gains they bring. The following statements reflect this indicator.

(…) by the informal feedback that the users and staff of the day centers give us verbally, such as their satisfaction at taking part in this activity and the fact that they feel more agile, feel that these exercises help them feel better, and the way they welcome us at the start of the sessions.(P2)

We assess adherence to the programs using the following indicators: satisfaction with the Program, assessed by a satisfaction questionnaire; objectives achieved in the program; maintenance of the gains made in the program 6 to 12 months post-program.(P4)

They recognize that after its implementation they have made gains in their health and in their lives. We’ve had cases where, after the programs, they were able to return to activities that had been suspended for some time, such as swimming or even recovering some of their social life.(P6)

We have also seen enthusiasm and adherence through people’s involvement in other activities related to the project.(P8)

Patients say they feel their functionality improving day by day.(P8)


**Involvement**


The involvement of participants by their attendance or participation in the program sessions is expressed in the following statements.

We assess adherence to the programs using the following indicators: frequency of participation; duration of participation in the sessions; level of effort put into the sessions; (…), maintenance of the post-program exercise training plan.(P4)

I see it in a very positive light in that there is perhaps a greater difficulty in influencing to start the program, that is, the first session. Afterward, the continuity of the program is carried out and accepted in a fluid way, with no dropouts. On the contrary, even after being discharged from the clinic and, in turn, from the gym, many people show an interest in continuing with the program, even if it’s with a lower weekly or even monthly frequency in some cases.(P7)

We can verify adherence to the program by the fact that users come to the sessions. They have to come to at least 75% of the planned sessions.(P8)

Through the number of sessions, the person attends, as well as their involvement in the sessions.(P3)

Through the attendance register (…)(P2)

### 3.4. Health Gains from Implementing These Strategies

This category only includes the indicator “Promoting active and healthy aging”.


**Promoting active and healthy aging**


The promotion of active and healthy aging is evident in physical, psychological, and social well-being. The different assertions reflect these health gains resulting from participation in physical exercise programs, oriented towards a rehabilitation perspective.

On a physical level, there was a reduction in musculoskeletal pain, an increase in joint mobility, and an improvement in balance and motor coordination. On a psychological and mental level, they reported an improvement in their well-being and perception of health (results of the evaluation questionnaire). On a social level, they reported an increase and improvement in their social life due to their integration into the group and the expansion of their relationships.(P1)

Prevention of the complications of the aging process, people’s quality of life, reduction in falls, independence in the performance of their activities of daily living, and greater tolerance to exertion.(P2)

Lower risk of falls, lower risk of signs/symptoms of anxiety and depression; physical and mental well-being.(P3)

Implementing strategies to improve adherence to fitness programs leads to health benefits, reducing the progression of chronic respiratory diseases, improving cardiovascular and mental health, reducing the risk of falls and injuries, improving health-related quality of life, and reducing healthcare costs.(P4)

Functional and instrumental autonomy, improved quality of life, reduced dependence on others, and reduced costs associated with performing activities of daily living (namely reduced need for diapers or dependence on formal caregivers).(P5)

There have been gains in strength, agility, cardiovascular capacity, flexibility, reduced cardiovascular risk, improved performance of activities of daily living, improved quality of life, and fewer depressive/anxious symptoms.(P8)

Reduced symptoms and dyspnea, better muscle function, reduced muscle fatigue, better respiratory function, increased quality of life, better tolerance to exertion.(P6)

The health gains that we have been notifying are those related to improvement in the feeling of tiredness; greater functional capacity; more muscular capacity; greater tolerance to daily life activities; greater capacity for physical exercise; change in lifestyle with the incorporation of Physical Exercise into their daily routines; in some cases a reduction in pain complaints; improvement in quality of life (in the broadest sense).(P7)

The reduction in the number of consultations and hospitalizations due to exacerbations of the disease, the correct use of inhalers, the improvement in quality of life, and the risk of anxiety and depression…(P9)

### 3.5. Gains from Group Activities

This category includes five indicators: “Health promotion”, “Health education”, “Active aging”, “Social interaction” and “Motivation for the program”.


**Health promotion**


Group participation in physical exercise programs is a promoter of physical and mental health, as expressed in the following recording unit.

(…) this group program has a positive influence on a healthier life, both physically and mentally.(P2)


**Health education**


Group activities facilitate involvement and learning, particularly in physical exercise programs where the educational component is present.

(…) group health education (…)(P5)


**Active aging**


Active aging is one of the benefits of participating in group activities, which is reflected in physical, psychological, and social well-being. The following statement expresses this dimension.

Decreased isolation, increased quality of life, improved physical condition and mood, decreased risk of vulnerability, and decreased risk of falling.(P1)


**Social interaction**


Participation in group physical exercise programs is crucial for older people’s mental health because they share their feelings, difficulties, and experiences, because they feel they belong to a group because they socialize, and because of the help and support they receive during their involvement and the dynamics that are built up during group activities. The following statements express these gains.

Sharing feelings, difficulties, and mutual encouragement to practice physical exercise.(P2)

Sense of belonging.(P3)

Group practice can be a form of companionship and fun, making physical activity more enjoyable and less monotonous.(P4)

Social interaction, which could be beneficial for participants’ mental and emotional health.(P4)

(…) the possibility of exchanging experiences between participants about their experiences of illness and rehabilitation.(P5)

(…) socialization, getting out of the house, maintaining a routine (…)(P6)

Socially, it’s more productive because, through contact with other people from similar age groups, they can expose some difficulties and share strategies for improvement; it can overcome the feeling of loneliness and promote relaxation in future sessions.(P7)

This supportive relationship also allows many of them to break out of isolation and have some kind of support network.(P8)

Carrying out this type of group activity allows experiences and strategies to be shared that can help people going through the same type of problem/situation, even those whose chronic pathologies are not yet causing acute symptoms (hypertension, CHF, diabetes, COPD, etc.).(P9)


**Motivation for the program**


One of the benefits of participating in group exercise programs is the motivation of the participants. The following statements reflect these gains.

Motivation based on the feedback they get from other people.(P3)

Motivation, through mutual support and encouragement between participants.(P4)

Motivation, (…)(P5)

Motivation, (…)(P6)

(…) and possibly greater adherence to exercise.(P6)

(…) the “herd effect” in which the participants influence each other in some way, stimulating each other and acquiring a kind of healthy competition (…)(P7)

Yes, the elderly in a group are more involved, they stimulate each other to achieve their goals. You can see that the group spirit even influences the results.(P8)

## 4. Discussion

The physical exercise program for the older person in the community was highlighted in terms of its purpose, evaluation, and components.

Aging is linked to lower cardiorespiratory fitness and muscle function, with all the repercussions for functional independence and the quality of life of the older person. Structured physical exercise programs, with the intervention of a rehabilitation nurse, help to mitigate these changes. They are a decisive strategy in the prevention of many chronic diseases or their worsening, and in improving mobility, cognition, and the functionality and well-being of older people [4,11].

Physical exercise should be planned, structured, and based on the outcome to be achieved. This means that it should be adjusted to the needs of each person and should produce an appropriate response depending on the different components [4]. Programs coordinated by rehabilitation nurse specialists can involve motor and respiratory re-education, duly aligned with a rehabilitation philosophy, as highlighted by the participants in this study.

As for the components, the type of program varies. In addition to warm-up exercises at the beginning of the program and relaxation exercises at the end, the types of program used by these health professionals have been adjusted to the capabilities of the older person, in line with international recommendations, i.e., taking into account different conditioning factors, such as health history, musculoskeletal limitations, functional capacity, tolerance to effort, and personal preferences [4]. They aim to maintain or improve one or more components of the elderly person’s physical capacity, such as muscle strength, joint mobility, endurance, balance, flexibility, and aerobic training [4].

A multicomponent exercise program involving aerobic training, strength training, flexibility training, balance and coordination training, as well as activities of daily living training, is essential for maintaining and improving functional capacity and preventing falls among the older person [4], in line with what has been found in several studies [7,8,9,13,14,16].

Although the sequences of the exercise program are not explicit in the rehabilitation nurses’ interviews, international recommendations point to the introduction of muscle strength training, then balance exercises, and finally aerobic training. This typology is seen as the key to the success of multicomponent exercise programs [4].

Warm-up exercises were highlighted in the discourse of these rehabilitation nurses, given their importance for the cardiovascular and musculoskeletal systems. They involve performing exercises at a lower intensity. They should begin with a single type of exercise, with a view to the adherence and gradual adaptation of the elderly person to the different components of the program [4].

In terms of frequency and time (duration), the discourses of these participants state predominantly between two and three times a week and with a time of approximately 1 h, as we can find in some studies [8]. The duration/time varies approximately between 16 and 20 sessions. The programming of these components is in line with international recommendations [4], which advocate adjusting them to the physical fitness and confidence of the elderly person. The assessment of heart rate and subjective perception of exertion, through the application of the modified Borg scale, provides valuable information for adjusting the frequency, duration, and intensity of the program to the capabilities of the elderly person, particularly with regard to their adaptations regarding aerobic fitness, gait, and mobility [4].

In the narrative review developed by Haider et al. [10], involving randomized controlled trials focused on frail and pre-frail elderly people living in the community, the studies analyzed differed in various aspects of the exercise programs used and the methods of intervention (health professionals, volunteers, at home, or in health institutions). The results point to the effectiveness of the interventions in reducing frailty and increasing muscle strength and physical performance.

The randomized trial developed by Rezola-Pardo et al. [16] aimed to compare the effects of a multicomponent exercise intervention and a walking intervention on physical and cognitive performance, habitual physical activity, affective function, and quality of life among older people (81) living in nine residential care facilities. The multicomponent exercise group took part in an individualized and progressive two-week program involving strength and balance exercises over 3 months, while the group that took part in the walking intervention was also individualized and participants walked 20 min a day. In terms of results, the group that took part in the multicomponent exercise showed greater improvements in physical performance. There were no significant differences in cognitive performance or habitual physical activity and both groups showed improvements in anxiety and quality of life. This study highlighted the importance of interventions involving multicomponent exercise [16].

The study by Mulasso et al. [14], concluded that multicomponent exercise programs can be effective in improving and reversing frailty, specifically for frail and pre-frail people. One hundred and twenty-three community-dwelling older adults were involved, with the experimental group comprising 62 participants and the control group 61. The experimental group took part in a 16-week exercise program involving endurance, strength, balance, and flexibility exercises, while the control group kept to their routine. It was found that after the exercise program, the experimental group showed greater strength than the control group. The effects of the training were greater in frail and pre-frail people (reductions in frailty of 0.67 and 0.76 points, respectively) compared to robust people (whose levels of frailty increased by 0.23 points; F = 11.32, *p* < 0.001) [14].

The study by Makizako et al. [13] also made it possible to verify the effects of a multicomponent exercise program on the physical function of elderly people with sarcopenia. This was a randomized, blinded, controlled clinical trial which involved 72 elderly people with sarcopenia or pre-sarcopenia. Participants were randomly assigned to the exercise and control groups. The program lasted 12 weeks, with 60 min per session, and included structured resistance exercises, and balance, flexibility, and aerobic training. The evaluation focused on physical function and muscle mass, and it was found that the exercise program essentially improved physical function in the intervention group, while it was not clear whether the program was effective in increasing muscle mass [13]. Multicomponent exercise programs for elderly people, in line with what has been found in different studies [10,13,14,16], can be developed in groups, in the community, or individually at home, with direct intervention from rehabilitation nurses or others health professionals, and are aimed at reducing frailty and maintaining their functional independence.

The physical exercise programs developed by rehabilitation nurses for older people in the community have been decisive in promoting health, functional independence, well-being, and self-care. Physical exercise is a preventive measure, along with other factors which favor a healthy lifestyle and prevent multiple health problems [4]. The prevention of falls has become one of the concerns of these nurses, and in this sense, the combination of balance and resistance training is fundamental to reducing falls in the elderly [23].

It is necessary to maintain the ability to perform physical exercise to prevent significant health consequences, such as the ability to perform activities of daily living and maintain independence [24]. The perspective of these participants regarding the importance of physical exercise programs is in line with what the WHO [6] advocates, that healthy aging presupposes the maintenance of functional capacity and quality of life.

Multicomponent physical exercise programs are fundamental in promoting health and quality of life in the elderly. The experimental study by Leitão et al. [25] demonstrates the relevance of physical training on the hemodynamic and lipid profile of elderly women with hypertension and dyslipidemia. Nineteen elderly hypertensive women with dyslipidemia were selected for the exercise group who underwent the multicomponent program for nine months, followed by a period of one year without training, while the control group maintained their daily routine without exercise. It was found that the multicomponent exercise program significantly improved the hemodynamic and lipid profiles, as well as the functional capacities of these elderly, hypertensive women with dyslipidemia. Although a period without training is detrimental to these benefits (in systolic blood pressure, diastolic blood pressure, resting heart rate, glycemia, total cholesterol, triglycerides, agility, lower and upper limb strength, cardiorespiratory capacity, and lower and upper limb flexibility), the first three months without training are most prominent in relation to these changes (with the exception of diastolic blood pressure) [25].

The study developed by Faria et al. [8] aimed to evaluate the effects of a rehabilitation nursing program on the functional capacity and lifestyles of frail elderly people. It was a randomized, controlled, two-group clinical trial with 30 frail elderly people enrolled in a health unit in Portugal between 2021 and 2022. The exercise program lasted 12 weeks and the sessions took place in the homes of the elderly participants. After the program, there was an improvement in multidimensional and physical frailty, functional capacity, balance, and perceived exertion in the experimental group. In addition, there were significant improvements in the physical activity habits, relational behavior, and stress management of these elderly people. The reality that is implicit in this study [8] should be part of everyday healthcare for elderly people in the community experiencing frailty. It is essential to integrate nurses with specialized training in the area of rehabilitation into community health teams, to offer an effective response to elderly people with this type of problem.

Functional independence was the central objective of Mendes’s quasi-experimental study [12], which compared the functional independence of two groups of elderly people with initially similar characteristics, living in a sedentary residential structure for the elderly. One of the groups underwent a physical exercise program, which included muscle strength training and aerobic training, for 12 weeks, 5 days a week, at a moderate intensity. The Functional Independence Measure (FIM) was used to assess the functional independence of the elderly before the program and at a second point, i.e., after the program had been implemented in the experimental group. It was found that the experimental group improved their functional independence in the “bath”, “bath and shower” and “stairs” sub-levels, while maintaining all the others, and proved the importance of physical exercise in maintaining and/or improving the functional independence of the elderly.

Older people’s adherence to exercise programs was related to their expressions of satisfaction with the programs or to the gains that resulted from their participation, as well as their involvement in physical exercise programs. The strategies used by nurses to involve participants in the definition of objectives, in decisions regarding the training plan, and in promoting the motivation of older people to get involved and participate are a determining factor in their adherence to the program [8]. On the other hand, the greater independence of community dwellers (noninstitutionalized) and their motivation to participate in these rehabilitation programs are determinants of their adherence to the interventions [10]. It should be noted that in the study by Gouveia, et al. [9] there was a high adherence rate (100%), evidenced by the participation of elderly people in the intervention group. These data revealed a high level of acceptance and motivation on the part of the participants with regard to the intervention program. The results of the study by Gouveia et al. [9] regarding the adherence of older people to participation programs are consistent with the perspectives of the rehabilitation nurses in this study.

The promotion of active and healthy aging emerges as health gains from the implementation of these rehabilitation strategies for the elderly. This perspective is in line with international recommendations for exercise in older adults to prevent diseases at all three levels of prevention [4]. Participation in structured exercise programs offers gains against a wide range of diseases and disabilities associated with aging. Programs can be individualized and controlled to achieve the desired result, with necessary adaptations in the different components of the programs [4].

Continuous rehabilitation at home can clearly improve quality of life and participation in activities of daily living for older people. This is the main conclusion of the study carried out by Imanishi et al. [11] and fits in with the health gains that emerged in this research. In the nonrandomized controlled intervention trial carried out by Imanishi et al. [11], the elderly participants were separated into the rehabilitation group and the nonrehabilitation group, with 100 elderly people in each. The rehabilitation group received a physical treatment program once a week and basic nursing care at home, which included assistance with cooking, cleaning, hygiene, meals, and medication. The nonrehabilitation group received only basic nursing care at home. In each group, quality of life and activities of daily living were assessed every three months for a period of one year. In the results, the rehabilitation group showed statistically significant improvements in quality of life and activities of daily living compared to the nonrehabilitation group.

The study by Pires et al. [15], also showed significant health gains for the elderly, improving balance, pelvic floor muscle competence, and cognitive performance, with a better quality of life for the participants. In this quasi-experimental, before—after study with a control group, 30 elderly people took part, 16 in the intervention group and 14 in the control group. It involved a 12-week program for active, institutionalized elderly people, with two-weekly sessions of balance and pelvic floor muscle training, with once-weekly sessions of cognitive stimulation interspersed. The people who took part in the intervention group improved in static and dynamic balance, fear of falling, and gait execution. They showed benefits in cognition and perceived quality of life associated with urinary incontinence [15]. The results of studies involving institutionalized elderly people [12,15] highlight the importance of residential structures for elderly people developing health promotion strategies for their clients, which must inevitably include physical exercise programs adjusted to the needs of elderly people.

Social interaction stands out when participating in group exercise programs because of its importance for older people’s mental health and the influence the group has on the motivation of its participants. Participation in physical exercise programs has led to greater social participation and contributes to improving the health and well-being of these elderly people. These findings are consistent with those of Fain et al. [26], who consider that physical activity modifies the effects of social participation on mortality, as evidenced by the association between social participation and lower mortality in older people. These findings are also in line with the study by Rocha et al. [17], who concluded that sociability was one of the main motivational dimensions which led older people to take part in physical activity, showing the importance of taking part in physical activity for social interaction and building bonds for older people.

Participation in physical exercise programs is one of the challenges of the future as a mandatory part of health care for the elderly, whether in a community setting, in residential facilities for the elderly, or in a hospital environment [4].

This research has limitations related to the method, the intentional selection of participants, and the data collection technique. The semi-structured interview script allows for some flexibility and increases the richness and depth of the findings, but increases the difficulty in conducting the interviews and influences the diversity of responses. The use of a researcher with experience in conducting semi-structured interviews and developing qualitative studies may have been important in minimizing this risk of bias. Future studies with a larger number of participants should delve deeper into strategies that enable the development of structured physical exercise programs in a community context.

This study can make an important contribution towards reflecting on this issue and the relevance of specialized rehabilitation nursing interventions, with a view to implementing effective strategies which ensure an active and healthy life for people who are aging [27]. These evidence-based strategies integrate structured physical exercise programs, adjusted to the care needs of older people, in order to maintain and improve their physical fitness, mobility, and functional capacity [3,28], and prevent some geriatric syndromes such as falls [29,30], immobility and even cognitive decline [31], as regular exercise can positively influence cognitive ability, reduce the rate of cognitive aging, and even reduce the risk of dementia [32].

The implementation of programs that integrate physical activity and physical exercise may contribute to active aging, with gains in functionality and quality of life, as recommended by the World Health Organization [6]. In this sense, it is essential that political decision-makers at the local level plan strategies that allow the development of programs aimed at the functional independence of elderly people in the community and their rehabilitation. Health policies must value the intervention of nurses and/or other health professionals in the rehabilitation area to intervene with multicomponent exercise programs and assume them as an integral part of health care in the rehabilitation of elderly people.

## 5. Conclusions

Nurses specializing in rehabilitation nursing recognize the importance of a structured physical exercise program aimed at the preferences and characteristics of older people. There is evidence in the nurses’ interviews regarding how the programs are structured, following the acronym FITT-VP, describing elements such as frequency, intensity (safety), time, type of exercise, volume, and progression. The reasons for the importance (advantages) and the indicators relating to the adherence of older people to the programs are identified. Finally, the individual and group health gains resulting from the implementation of the programs are described.

The design and implementation of physical exercise programs which are structured and adapted to the circumstances of older people bring advantages in rehabilitation and the maintenance of active aging, with benefits in terms of well-being and functionality, and consequently in terms of quality of life.

## Figures and Tables

**Table 1 jfmk-08-00163-t001:** Categories and indicators, 2023.

Category		Indicator
**Physical exercise program for the** **older person** **in the community**	Purpose and evaluation	– Program definition – Program objective – Person assessment – Assessment tools – Conditioning factor
	Components	– Intensity of the program – Frequency and timing – Type of program – Volume and progression – Educational component
**Importance of physical exercise in the rehabilitation of the** **older person** **in the community**		– Health promotion – Functional capacity – Well-being and self-care
**People’s adherence to the programs**		– Feedback and satisfaction – Involvement
**Health gains from implementing these strategies**		– Promoting active and healthy aging
**Gains from group activities**		– Health promotion – Health education – Active aging – Social interaction – Motivation for the program

## Data Availability

The data presented in this study are available on request from the first author.

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
