# Peer review of "The Perspective of Rehabilitation Nurses on Physical Exercise in the Rehabilitation of Older People in the Community: A Qualitative Study"

_jfmk, 2023, doi:10.3390/jfmk8040163_

Round 1
Reviewer 1 Report
Comments and Suggestions for Authors
The authors present a study underscoring the importance of structured exercise programs for the elderly, focusing on the perspective of Rehabilitation Nurse Specialists (RNS). Given the global aging population, this study gains significance. It emphasizes the distinction between physical activity and structured exercise, providing essential parameters for program design. Recognizing the vital role of RNS in program development, the study advocates for a multidisciplinary approach. These insights hold implications for optimizing elderly care. However, I would recommend a major revision for conciseness.
Title has two sentences, please revise, and make a reflective title.
Line 29; 5 caterories? Are the author implying themes, if not please explain what categories is being referred. The abstract need to revised significantly to improve
While the introduction provides a thorough background, it could benefit from a more explicit statement of the study's specific objectives and research questions.
While the introduction outlines the importance of implementing physical exercise programs for the elderly, it could benefit from explicitly stating the research gap or the specific contribution this study aims to make to the existing body of knowledge.
There is some repetition of information, particularly in the discussion of terms like "physical activity" and "physical exercise." Try to consolidate and streamline the explanations for a more concise and focused introduction.
Methods
Section 2.3: please have the interview guide a supplementary material
How was the study data analysed, the methods should clearly describe. Please review this recent paper which has major key aspect of qualitatiove research which can be covered and explained https://www.mdpi.com/2227-9032/11/19/2665
what was analysis approach ? thematic analysis? Grounded theory. The study has a poor rigor.
Please have a section on rigor as well.
Please focus on methodical aspect on methods, and I believe authors can have a short brief paragraph on ethics rather than multiple paragraphs.
Result
Please revisit the results, as the readability of the paper is very poor in its current version. Some improvements could be made by limiting the number of quotes, such as those in lines 406-425. They could potentially be transformed into a table or limiting to best quotes..
Some of the used quotes are not clear. Examples can be found in lines 429, 487, and 544. These are just a few instances of many. Please ensure that statements from the interview are complete and understandable.
Discussion:
The discussion covers a wide range of information from various studies. Consider organizing the information into more defined paragraphs to make it easier for the reader to follow the flow of arguments.
It would be beneficial to explicitly state how the findings of the studies discussed can be applied in practical settings, especially for healthcare professionals and policy-makers.
While the discussion predominantly focuses on the positive effects of structured exercise programs for the elderly, consider briefly addressing some challenges associated with such programs.
it would be useful to briefly touch on the methodologies employed in these studies. This could provide context for the credibility and reliability of the findings.
potential areas for future research in this field can be embedded in the discussion.
Comments on the Quality of English Language
The paper must be revised significantly.
Author Response
Reviewer 1
Thank you very much for taking the time to review this manuscript. Please find the detailed responses below and the corresponding revisions/corrections highlighted in the documents that follow.
|
Comment 1: Title has two sentences, please revise, and make a reflective title. |
|
Answer 1: Thank you for your suggestion. We agree with this comment, and we have changed the title which traduces the content and the method of this study. This change can be found on page 1 of the article, line 3 to 5. |
|
|
|
Comment 2: Line 29; 5 categories? If the author is implying themes, if not please explain which categories are being referred to. |
|
Answer 2: Thank you for your comment. We agree that the presentation of results should be clear and concise. We have made changes to the abstract on page 1, line 29 to 35. |
|
|
|
Comment 3: Although the introduction provides a thorough background, it could benefit from a more explicit statement of the study's specific objectives and research questions. |
|
Answer 3: Thank you for your suggestion. We agree with this comment and have chosen to introduce three specific objectives that allow the object of study to be measured more concisely. This change can be found on page 3, line 102 to 110. |
|
|
|
Comment 4: Although the introduction outlines the importance of implementing exercise programs for the elderly, it could benefit from making explicit the research gap or the specific contribution this study intends to make to the existing body of knowledge. |
|
Answer 4: Thank you for your suggestion. We agree with the comment and have decided to introduce a paragraph explaining the contribution of this study to the field of science and specialization. This change is expressed on page 3, from line 111 to 116. |
|
|
|
Comment 5: There is some repetition of information, particularly in the discussion of terms such as "physical activity" and "physical exercise". Try to consolidate and streamline the explanations for a more concise and focused introduction. |
|
Answer 5: Thank you for your comment on the terms "physical activity and physical exercise". We agree with the need for a concise and focused approach. In this sense, we have taken care to differentiate the terms physical activity and physical exercise, using the "International Exercise Recommendations in Older Adults (ICFSR): Expert Consensus Guidelines" (2021) and in agreement with the suggestions made, we have taken care to use the term "exercise program" whenever we characterize physical exercise as a planned and structured activity. This change can be found on page 2, line 77 up to page 3, on line 81 and then between lines 85 and 87 on page 3. |
|
|
|
Comment 6: Section 2.3: please have supplementary material in the interview guide material. How the study data was analyzed, the methods should clearly describe. Please review this recent article which has an important key aspect of qualitative research that can be covered and explained https://www.mdpi.com/2227-9032/11/19/2665. What was the analysis approach? Thematic analysis? Grounded theory? |
|
Answer 6: Thank you for your suggestions regarding data analysis. We agree with the concern to make the analysis approach used explicit in the article. The answer to this question is explained in section 2.4 (Data processing and analysis). We used thematic analysis, along the lines of Laurence Bardin, a French author whose original work on content analysis was published in 1977. On page 5, from line 164 to line 176, we took care to concisely present the data analysis procedures used, using content analysis, more specifically thematic analysis. In this process, we were concerned to present the corpus of analysis, the coding units, with the necessary explanation of what is meant by theme according to Laurence Bardin, given that we are dealing with a thematic analysis, in addition to the process of categorization. We agree that the categorization process needs to be better explained and so we have supplemented the information introduced, the change to which can be seen on page 5, lines 176 and 179. |
|
|
|
Comment 7: The study has poor rigor. Please have a section on rigor as well. |
|
Answer 7: Thank you for your suggestions regarding rigor. Although the end of section 2.4 deals with the rigor of research, we agree with this comment and have therefore created section 2.5 referent to Rigor and Trustworthiness of the Research, which responds more concisely to the criteria of rigor and quality of qualitative research. This change is visible on page 5 of the revised manuscript, from line 182 to 198. |
|
|
|
Comment 8: Please focus on the methodical aspect about methods, and I believe authors can have a short paragraph about ethics instead of several paragraphs. |
|
Answer 8: Thank you for your comments. We agree with the need to be methodical in the approach to ethical aspects and to explicitly highlight the different procedures that lend credibility to the concerns about the ethical aspects of this study. With the changes made to the article, the ethical aspects correspond to section 2.6. |
|
|
|
Comment 9: Please review the results, as the readability of the article is very poor in its current version. Some improvements could be made by limiting the number of quotes, such as those in lines 406-425. They could potentially be turned into a table or limited to the best quotes. Some of the quotes used are unclear. Examples can be found in lines 429, 487 and 544. These are just a few examples of many. Make sure the interview statements are complete and understandable. |
|
Answer 9: Thank you for your suggestions. We agree with this comment and have removed the registration units (the quotes referred to by the reviewer correspond to registration units/themes in Bardin's thematic analysis) that were not clear enough. What is marked in yellow corresponds to the text that has remained intact. |
|
|
|
Comment 10: The discussion covers a wide range of information from various studies. Consider organizing the information into more defined paragraphs to make it easier for the reader to follow the flow of arguments. It would be beneficial to make explicit how the findings of the studies discussed can be applied in practical settings, especially for health professionals and policy makers. Although the discussion focuses predominantly on the positive effects of structured exercise programs for the elderly, consider briefly addressing some of the challenges associated with such programs. It would be useful to briefly address the methodologies employed in these studies. This could provide context for the credibility and reliability of the findings. Potential areas for future research in this field could be incorporated into the discussion. |
|
Answer 10: Thank you for your suggestions. We agree that the discussion covers a wide range of information related to the categories defined in the analysis and the indicators that emerged in the thematic analysis (in the results). Most of the studies presented highlight the methodology used. It has been added on page 17, line 655, on page 18, line 699 and on page 19, line 722. The discussion very briefly addresses some of the challenges arising from this study, on page 20, line 788 to 790 and line 801 to 809. |

Reviewer 2 Report
Comments and Suggestions for Authors
The authors in the report claim that nurses have to design and implement programs aimed at the elderly in order to keep them healthy and maintain their quality of life. This is a well-known truth that should be also enriched with the statement that it is also a task for rehabilitation doctors and physiotherapists, actually well performed by them all over the world. The article includes more well-known statements that cannot be recognized as novelty e.g. Abstract, lines 21-23 ..." Structured physical exercise and overall physical activity play a major role in maintaining an active lifestyle, improving health, preventing disease, and helping to maintain older people's quality of life."... . Moreover, the aim of the study (lines 23-24) for Authors is..." To investigate the importance of implementing physical exercise programs for the elderly in the community, taking into account the perspective of the Rehabilitation Nurse Specialist. ... ." is difficult to estimate because such tasks are always faced by nursing specialists, so the question arises: what did the authors intend to investigate?
The methodology was based on questionnaires and reports with nine rehabilitation nurses from various regions of Portugal who have experience implementing physical exercise programs with the elderly, which seems to be subjective and free in evaluation as well as drawing conclusions. Five categories were studied: "Physical exercise program for the elderly in the community", "Importance of physical exercise in the rehabilitation of the elderly in the community", "People's adherence to the programs", "Health gains resulting from the implementation of these strategies" and "Gains from group activities" and the conclusions were easy to be predicted that (cit.) ..."Nurses recognize the importance of using structured physical exercise programs tailored to older people, bringing advantages in rehabilitation and maintaining active aging."... There is no rational explanation for data analysis accepted by the authors, based only on the expertise, which does not consider the inter-rater reliability.
The discussion is not very comprehensive, the research results are compared with those obtained in a few centers outside Portugal; only 27 refs. are provided.
Comments on the Quality of English LanguagePoor English quality
Author Response
Reviewer 2
Thank you very much for taking the time to review this manuscript. Please find the detailed responses below and the corresponding revisions/corrections highlighted in the documents that follow.
|
Comment 1: The authors of the report state that nurses have to design and implement programs aimed at the elderly in order to keep them healthy and maintain their quality of life. This is a well-known truth that should also be enriched with the statement that it is also a task for rehabilitation physicians and physiotherapists, in fact well performed by them all over the world. The article includes more well-known statements that cannot be recognized as novel, for example, Summary, lines 21-23 ..." Structured exercise and general physical activity play an important role in maintaining an active lifestyle, improving health, preventing disease and helping to maintain quality of life for older people." ... . Furthermore, the aim of the study (lines 23-24) for the authors is..." To investigate the importance of implementing physical exercise programs for the elderly in the community, taking into account the perspective of the Rehabilitation Specialist Nurse. is difficult to estimate, as such tasks are always faced by nursing specialists, the question arises: what did the authors intend to investigate? |
|
Answer 1: Thank you for your suggestion. We agree with your comment regarding the statement that nurses, doctors, and physiotherapists must design and implement programs in order to keep older person healthy and maintain their quality of life. However, the object of this study focuses on the intervention of rehabilitation nurses in relation to physical exercise programs in the rehabilitation of elderly people in the community. We can't ignore the fact that in Portugal there is a specialty area in nursing, rehabilitation nurses, whose professionals, due to their skills, have an intervention in a community context that is very appropriate and close to the elderly population. The work between the different professionals is one of partnership and complementarity, but in clinical contexts in the community, the development of these programs and projects is often in the direct responsibility of the nurses.
We agree with the comment regarding the objective and have decided to make a change in the abstract, with the reformulation of the aim of the study and in the introduction with the definition of specific objectives. This change can be found on page 1, lines 23 and 24 and page 3, lines 102 to 110. |
|
|
|
Comment 2: The methodology was based on questionnaires and reports with nine rehabilitation nurses from various regions of Portugal who have experience in implementing physical exercise programs with the elderly, which seems to be subjective and free in evaluation as well as in drawing conclusions. Five categories were studied: "Physical exercise program for the elderly in the community", "Importance of physical exercise in the rehabilitation of the elderly in the community", "People's adherence to the programs", "Health gains from implementing these strategies" and "Gains from group activities" and the conclusions were easy to predict that (cit.) ..."Nurses recognize the importance of using structured physical exercise programs adapted to the elderly, bringing advantages in the rehabilitation and maintenance of active aging". ... There is no rational explanation for the data analysis accepted by the authors, based solely on expertise, which does not consider inter-observer reliability. |
|
Answer 2: Thank you for your comment. We agree with the use of these participants with experience in implementing physical exercise programs with the elderly in the community. In this sense, we opted for a qualitative approach because we felt it would be more appropriate to understand the phenomenon under study, valuing the meaning of the experience lived by these participants and taking an interest in understanding the object of study from the perspective of the subjects involved. We didn't intend to carry out an extensive study, but rather an in-depth one. Hence, the criterion that determined the number of subjects was data/findings saturation. We should also emphasize that this study was conceived from a naturalistic and interpretivist perspective, whose scientific efforts were directed towards understanding the phenomenon under study, which corresponds to contextual knowledge. We assume that reality is multiple and that this object of study is unique, seen from the perspective of these nurses and not predictable, as is the case with a positivist approach. The results of the analysis were not based on the expertise of the researchers, seen in the pejorative sense of the term. They are the reflection of rigorous, high-quality work, compatible with what is recommended in qualitative research and in line with the criteria of credibility, transferability, dependability, and confirmability. A change was made to the article, with the creation of section 2.5 about Rigor and Trustworthiness of the Research, visible on page 5 of the revised manuscript, from line 182 to 198. |
|
|
|
Comment 3: The discussion is not very comprehensive; the research results are compared with those obtained in some centers outside Portugal; only 27 refs. are provided. |
|
Answer 3: Thank you for your suggestion. We believe that the discussion covers a wide range of information related to the categories defined in the analysis and the indicators that emerged in the thematic analysis (in the results). We used studies carried out by researchers from the scientific area in question and from other areas, with similar skills (some from Portugal and others from other countries). With regard to the number of references, we tried to use as many authors and studies as possible in order to lend credibility to the interpretation of the findings that emerged from the analysis process. As a result of the improved discussion, two more references were added, bringing the total to 29. This change can be seen on page 23, lines 908 to 912. |
|
|
|
Comment 4: Poor quality English |
|
Answer 4: Thank you for your comment. Although AJE's highly qualified native English-speaking editors are involved in the translation process, we will endeavor to improve the quality of the text. |

Round 2
Reviewer 1 Report
Comments and Suggestions for Authors
The manuscript has been carefully reviewed, and I appreciate the effort put into this research. However, there are several concerns that need to be addressed before it can be considered for publication.
The response to comments, particularly in the methods and discussion sections, is minimal. It is crucial that the authors engage more substantially with the feedback provided, addressing each comment thoroughly and making appropriate revisions to the manuscript.
The manuscript currently lacks the methodological rigor expected for a scientific original research publication. At its current state, it reads more like an opinion or perspective piece rather than a research paper. I recommend revising and strengthening the methodology to ensure it meets the standards for original research.
Comments on the Quality of English Language
Required improvement.
Reviewer 2 Report
Comments and Suggestions for Authors
The authors have significantly adapted the step-by-step text of the manuscript to address most of my criticisms. This concerns the reformulation of the content in the Abstract, the state-of-art in the Introduction of the paper, and the description of the questionnaire research and review methodology. However, the methodology did not change. Furthermore, the authors explained the importance of nursing care in the form of rehabilitation exercises for elder people in Portugal, which slightly strengthened the belief in the scientific value of the paper. The Discussion of the manuscript has been extended and reworded with the addition of two additional references.
To sum up - The work has been corrected to the level almost necessary for publication.
Round 3
Reviewer 1 Report
Comments and Suggestions for Authors
The authors have responded minimally to the comments, particularly in the methods and discussion sections, and have made minimal changes to the manuscript.
Comments on the Quality of English LanguageModerate editing of English language required
Author Response
Dear Reviewer
Thank you very much for taking the time to review this manuscript. Please find the detailed responses below and the corresponding revisions/corrections highlighted in the documents that follow.
|
Comment 1: The authors responded minimally to comments, particularly in the methods and discussion sections, and made minimal changes to the manuscript. |
|
Answer 1: We appreciate the suggestions regarding methodology and discussion. We agree with the concern to improve the methodology. Therefore, we tried to make the data analysis process clearer and more complete. These changes are visible on page 6, between lines 213 to 219. In the discussion of the data and in order to frame the results of the studies discussed with practical environments, we introduced changes on page 19, line 726 to 730, on page 20, line 765 to 769 and on page 21, line 824 to 827. We briefly discuss some challenges associated with these physical exercise programs on page 22, line 859 to 867. We added some references, visible on page 22, line 855 to 858 and on pages 24 and 25, lines 971 to 978. |
|
|
Best regards
Reviewer 2 Report
Comments and Suggestions for Authors
Accept the text in its present form. Some minor corrections in the literature list are necessary as marked in yellow in the attached pdf file.

Author Response
Dear Reviewer
Thank you very much for taking the time to review this manuscript. Please find the detailed responses below and the corresponding revisions/corrections highlighted in the documents that follow.
Best regards
|
Comment 2: Accept the text in its current form. Some minor corrections to the literature list are necessary, as marked in yellow in the attached pdf file. |
|
Answer 1: We appreciate your suggestions for improving references. We agree with the need to present references. The changes are visible on pages 23 to 25, from lines 903 to line 978. |
Round 4
Reviewer 1 Report
Comments and Suggestions for Authors
The authors has revised the manuscript and improved the quality of their methodological soundness.
Comments on the Quality of English LanguageModerate editing of English language required